# The Responsiveness of Breast Cancer Cells to Varied Levels of Vitamin B12, Cisplatin, and G-CSF

**DOI:** 10.3390/ijms26189086

**Published:** 2025-09-18

**Authors:** Volkan Aslan, Duygu Deniz Usta, Atiye Seda Yar Sağlam, Ahmet Özet, Osman Sütcüoglu, Kürşat Dikmen, Nuriye Özdemir

**Affiliations:** 1Department of Medical Oncology, Faculty of Medicine, Gazi University, 06560 Ankara, Türkiye; dr.volcanaslan@gmail.com (V.A.); ahmetozet@gmail.com (A.Ö.); sutcuogluo@gmail.com (O.S.); nyozdemir@yahoo.com (N.Ö.); 2Department of Medical Biology and Genetics, Faculty of Medicine, Gazi University, 06560 Ankara, Türkiye; atiyeseda@yahoo.com; 3Department of General Surgery, Faculty of Medicine, Gazi University, 06560 Ankara, Türkiye; kursatdikmen@yahoo.com

**Keywords:** breast cancer, cisplatin resistance, vitamin B12, G-CSF

## Abstract

Supportive agents, such as vitamin B12 (cobalamin, B12) and granulocyte colony-stimulating factor (G-CSF), are widely used during chemotherapy; however, their direct effects on tumor biology are not well understood. We evaluated the impact of pharmacological B12 and G-CSF, alone or in combination with cisplatin, on hormone receptor-positive (MCF-7) and triple-negative (MDA-MB-231) breast cancer cells, conducting in vitro assays of cell viability, cytotoxicity, caspase activation, mitochondrial membrane potential, and cytolytic protein expression. Neither B12 nor G-CSF alone induced cytotoxicity; instead, both promoted proliferation in a dose- and time-dependent manner. When combined with cisplatin, they consistently attenuated drug-induced cytotoxicity, suppressed caspase-3/-8/-9 activation, preserved mitochondrial integrity, and reduced perforin/granzyme expression, exhibiting stronger effects in MCF-7 cells. G-CSF markedly increased proliferation (>130% at 50 ng/mL), while B12 modestly enhanced viability and mitigated cisplatin-induced damage, particularly in triple-negative cells. These findings indicate that B12 and G-CSF can impair cisplatin efficacy by blunting apoptotic signaling and mitochondrial injury in different breast cancer subtypes. These preclinical findings warrant prospective, biomarker-driven in vivo and clinical studies to delineate the clinical contexts in which B12 and G-CSF can be safely integrated into supportive care without compromising antitumor efficacy.

## 1. Introduction

Tumor cells have an increased demand for vitamin B12 (cobalamin, B12) because of their rapid proliferation and dependence on DNA synthesis and methylation pathways [1,2,3]. In the bloodstream, B12 is primarily transported by transcobalamins (TCNs), which facilitate its absorption and systemic distribution [4,5,6]. Several studies have demonstrated that TCNs are overexpressed in various malignancies, and this overexpression has been associated with tumor aggressiveness and poor clinical outcomes [7].

This overexpression may contribute to the elevated serum B12 levels observed in patients with solid tumors such as breast, lung, and colorectal cancers—even in the absence of exogenous supplementation. These findings suggest that tumor-derived TCN overexpression may drive circulating B12 elevation and potentially enhance tumor progression [7,8,9,10,11]. Nonetheless, the precise role of B12 in tumor biology remains unclear, especially in the context of its impact on treatment response and apoptotic signaling in cancer cells [4,5,12].

Breast cancer represents a heterogeneous group of diseases encompassing multiple molecular subtypes, each with distinct biological characteristics and therapeutic vulnerabilities. Hormone receptor-positive (HR+) breast cancer, modeled using the MCF-7 cell line, and triple-negative breast cancer (TNBC), modeled using MDA-MB-231 cells, are among the most studied subtypes. While HR+ tumors typically respond well to endocrine therapies, TNBC lacks estrogen, progesterone, and human epidermal growth factor receptor 2 (HER2) expression and is characterized by a more aggressive clinical course with limited targeted treatment options. Systemic chemotherapy remains a cornerstone of therapy for both subtypes, and platinum-based agents such as cisplatin are frequently employed in TNBC and occasionally investigated in HR+ [13].

G-CSF is commonly used in oncology to prevent or manage chemotherapy-induced neutropenia. Interestingly, recent studies suggest that some tumors, including TNBC, may endogenously produce G-CSF, which may further modulate tumor behavior. Notably, the MDA-MB-231 cell line has been shown to express higher G-CSF levels compared to non-invasive breast cancer lines such as T47D and MCF-7. Elevated G-CSF expression has been linked to increased cancer cell migration and invasiveness [14,15].

In this study, we aimed to explore the potential effects of B12 and G-CSF on breast cancer cell proliferation and response to chemotherapy. Using the MCF-7 and MDA-MB-231 cell lines, we investigated whether exposure to these agents modulates cellular viability and the apoptotic response to cisplatin. Our objective was to determine whether commonly used supportive care agents may interfere with the efficacy of chemotherapy in biologically distinct breast cancer subtypes and to provide mechanistic insights into their potential contribution to therapeutic resistance.

## 2. Results

### 2.1. Assessment of How Single Doses of Vitamin B12, G-CSF, and Cisplatin Affect Cell Viability

The effects of B12, G-CSF, and cisplatin on the viability of MCF-7 and MDA-MB-231 cells were assessed by MTT after 24, 48, and 72 h of treatment with increasing doses of single agents (Figure 1A–C and Figure 2A–C). Increasing B12 dosages (0.0005–500 µM) did not affect a decrease in cell viability in either of the cell lines; in fact, some doses were shown to increase viability (Figure 1A and Figure 2A).

In MCF-7 cells, treatment with G-CSF (0.001–50 µg/mL) resulted in a significant dose- and time-dependent increase in viability (Figure 1B). No alterations were observed at 24 h; however, proliferation increased continuously at 48 and 72 h, particularly at concentrations ≥0.25 µg/mL. The greatest increase occurred at 50 µg/mL after 48 h, at which point viability exceeded 130% of the control, suggesting a robust mitogenic effect in luminal A breast cancer cells. Treatment with G-CSF (0.001–50 µg/mL) resulted in a strong dose- and time-dependent increase in MDA-MB-231 cell viability, which was observed at 48 and 72 h. (Figure 2B). The most pronounced effect was observed at 50 ng/mL after 48 h, at which point viability exceeded 120% of the control, suggesting a proliferative effect in TNBC cells.

Cisplatin treatment significantly decreased the viability of MCF-7 and MDA-MB-231 cells in a dose- and time-dependent manner (Figure 1C and Figure 2C). In MCF-7 cells, reductions became statistically significant at 5 µM, with maximal cytotoxicity (<20% viability) observed at 500 µM after 72 h. A significant decrease was observed following treatment at concentrations as low as 5 µM, with maximal cytotoxicity observed after 72 h at a concentration of 500 µM in MDA-MB-231 cells (*p* < 0.001). Compared to hormone receptor-positive models, the TNBC cells exhibited greater sensitivity to cisplatin, potentially due to the lack of receptor-mediated protective signaling and increased baseline replication stress. One-way ANOVA (analysis of variance) with post hoc testing confirmed these results (* *p* < 0.05; ** *p* < 0.01; *** *p* < 0.001), which are consistent with cisplatin’s known DNA crosslinking and apoptosis-inducing activity.

### 2.2. Effect of Dual and Triple Combinations of Cisplatin with Vitamin B12 and G-CSF on Cell Viability

The impact of dual and triple combinations of cisplatin with B12 and/or G-CSF on MCF-7 and MDA-MB-231 cell viability was evaluated across cisplatin concentrations of 20, 40, and 80 µM (Figure 3A,B). In the dual combination groups, co-administration of cisplatin with either B12 (10 or 50 nM) or G-CSF (10 or 50 ng/mL) attenuated cisplatin-induced loss of viability in both cell lines in a dose-dependent manner (Figure 3A). The greatest cytoprotective effect was observed at 40 µM cisplatin, with comparable restoration of viability at both low and high concentrations of the supportive agents, suggesting a plateau effect beyond a certain supplementation threshold. In the triple combination setting, where cisplatin was administered together with both B12 and G-CSF, partial recovery of viability was observed at lower cisplatin concentrations (20–40 µM), and the effect was more pronounced in MCF-7 cells compared to MDA-MB-231 cells (Figure 3B). At 80 µM cisplatin, the protective influence was markedly diminished, indicating a potential cytotoxic threshold above which these agents are unable to effectively counteract platinum-induced damage.

Although not all comparisons reached statistical significance, the overall trend indicated that B12 and G-CSF—either individually or in combination—consistently mitigated cisplatin-mediated cytotoxicity. These findings support the hypothesis that such supportive agents may inadvertently promote tumor cell survival and potentially compromise the therapeutic efficacy of platinum-based chemotherapy, particularly in tumor subtypes with intact apoptotic pathways.

### 2.3. Cytotoxic Effects of Cisplatin and Its Combinations with Vitamin B12 and G-CSF

Cytotoxicity was quantified by LDH release in the MCF-7 and MDA-MB-231 cells (Figure 4). Cisplatin (40 µM) alone produced a marked increase in cytotoxicity relative to the control—the result was more pronounced in MDA-MB-231 cells than in MCF-7 cells, indicating greater sensitivity of the triple-negative line. Co-treatment with B12 (50 nM) or G-CSF (50 ng/mL) lowered LDH release compared with cisplatin alone in both models, and the triple regimen (cisplatin + B12 + G-CSF) yielded the greatest attenuation, reducing cytotoxicity to roughly half of the cisplatin-only condition. The protective pattern was consistently stronger in MCF-7 cells than in MDA-MB-231 cells. Although not every comparison was statistically significant, the overall trend supports a cytoprotective effect of B12 and G-CSF against cisplatin-induced membrane damage, which aligns with their viability-preserving effects observed in companion assays.

### 2.4. Effects of Vitamin B12 and G-CSF on Cisplatin-Induced Caspase Activation

Caspase activity was quantified using a fluorometric multiplex caspase activity assay. Treatment with cisplatin (40 µM) alone markedly increased the activity of both initiator caspases (caspase-8 and caspase-9) and the executioner caspase (caspase-3) (Figure 5A–C). The magnitude of activation was generally greater in MDA-MB-231 cells compared to MCF-7 cells, suggesting a higher susceptibility of the triple-negative subtype to cisplatin. Co-treatment with B12 (50 nM) or G-CSF (50 ng/mL) significantly reduced the activation of all three caspases relative to cisplatin alone, while the triple combination (cisplatin + B12 + G-CSF) produced the most pronounced decrease, bringing activity levels close to those of the control group. These findings indicate that B12 and G-CSF attenuate cisplatin-induced activation of both the extrinsic caspase-8 and intrinsic caspase-9 apoptotic pathways, thereby limiting downstream caspase-3 activation, in line with their viability-preserving and cytoprotective effects observed in complementary assays.

### 2.5. Effects of Vitamin B12 and G-CSF on Cisplatin-Induced Loss of Mitochondrial Membrane Potential

Mitochondrial membrane potential (MMP) loss was evaluated using the TMRE (tetramethylrhodamine, ethyl ester) assay (Figure 6). Cisplatin (40 µM) treatment markedly increased MMP loss in both MCF-7 and MDA-MB-231 cells, with a greater effect observed in the MDA-MB-231 line. Co-treatment with B12 (50 nM) or G-CSF (50 ng/mL) significantly reduced cisplatin-induced MMP loss in both cell types. The triple combination of cisplatin, B12, and G-CSF had the most pronounced protective effect, restoring MMP values close to control levels. These results indicate that B12 and G-CSF help maintain mitochondrial integrity under cisplatin-induced stress, consistent with their inhibitory effects on caspase activation and overall cytotoxicity.

### 2.6. Effects of Vitamin B12 and G-CSF on Cisplatin-Induced Perforin and Granzyme Expression in MCF-7 and MDA-MB-231 Cells

A Western blot assay was performed to assess the effects of cisplatin, B12, and G-CSF on perforin, granzyme A, and granzyme B expression in MCF-7 and MDA-MB-231 cells (Figure 7). In MCF-7 cells, cisplatin (40 µM) treatment modestly increased the levels of all three proteins compared with the control group. B12 (50 nM) or G-CSF (50 ng/mL) alone caused minimal alterations, whereas their co-administration with cisplatin partially attenuated the cisplatin-induced increases. The triple combination (cisplatin + B12 + G-CSF) restored protein expression to values close to those of the control group. In MDA-MB-231 cells, cisplatin induced a more pronounced elevation of perforin, granzyme A, and granzyme B compared with the control. Similar to the findings in MCF-7 cells, treatment with B12 or G-CSF reduced these increases, and the triple combination exerted the strongest suppressive effect, restoring expression to baseline levels. These results are consistent with the reduction in caspase activation and the preservation of mitochondrial membrane potential shown in Figure 5 and Figure 6 (*p* < 0.05 and *p* < 0.001).

### 2.7. Combination Index Analysis

To further explore drug–drug interactions, MTT data were analyzed using the Chou–Talalay method (Figure 8). Across the tested effect range, interaction profiles differed by cell line and regimen. In MCF-7 cells, the triple regimen B12 + G-CSF + cisplatin was consistently antagonistic (CI, Combination Index ≈ 1.2–1.8 across Fa, fraction affected ≈ 0.15–0.40). Pairwise combinations performed better: B12 + cisplatin produced the lowest CI values, reaching additivity or mild synergy at select Fa points (CI ≈ 0.8–1.0 around Fa ≈ 0.25–0.45), whereas G-CSF + cisplatin was largely additive to mildly antagonistic, with only isolated CI values approaching 1. In MDA-MB-231 cells, the G-CSF + cisplatin combination was uniformly antagonistic over Fa ≈ 0.25–0.50 (CI ≈ 1.3–1.8), and no evidence of synergy was detected in this line under the tested conditions. Taken together, these data indicate that synergy is limited and context-dependent, emerging—when present—primarily for B12 + cisplatin in MCF-7 cells, while the triple regimen and the G-CSF-containing pair tend toward antagonism, particularly in MDA-MB-231 cels.

## 3. Discussion

Our in vitro findings demonstrate that B12 and G-CSF promote proliferation and enhance viability in both MCF-7 and MDA-MB-231 breast cancer cell lines. Neither agent alone induced cytotoxicity; however, they significantly reduced cisplatin-induced apoptosis and cytotoxicity when co-administered. This protective effect was most pronounced in MCF-7 cells, where B12 or G-CSF preserved mitochondrial membrane potential and decreased activation of caspases-3, -8, and -9, suggesting inhibition of both extrinsic and intrinsic apoptotic pathways. A similar, though slightly less marked, effect was observed in MDA-MB-231 cells.

The stronger cytoprotective response in MCF-7 cells may be attributable to the presence of estrogen receptor signaling and wild-type *p53,* both of which can modulate oxidative stress responses and apoptotic sensitivity [16,17]. Using both a luminal/ER-positive (MCF-7, wild-type *p53*) and a triple-negative/mesenchymal-like model (MDA-MB-231, mutant *p53*) provided complementary contexts in which to interpret the cytoprotective effects of B12 and G-CSF against cisplatin. In line with this rationale, the protective effects were more pronounced in MCF-7 cells, consistent with their distinct molecular features.

Collectively, these findings suggest that B12 and G-CSF may impair the pro-apoptotic efficacy of cisplatin across distinct molecular subtypes of breast cancer. Notably, this raises important translational concerns regarding the use of supportive care agents during cytotoxic chemotherapy, particularly in the case of aggressive or apoptosis-reliant tumors. Our Chou–Talalay analysis further confirmed that the protective effects of B12 and G-CSF translated into predominantly antagonistic interaction profiles with cisplatin, particularly in the MDA-MB-231 line, while synergy was observed only under limited conditions in the MCF-7 line.

From a clinical perspective, multiple population-based and integrative reviews report that supraphysiologic or persistently elevated plasma B12 is associated with a higher short-term incidence of newly diagnosed cancer and with increased mortality, although concerns regarding reverse causality and confounding (e.g., liver disease, occult malignancy) remain; these data argue against unnecessary supplementation in oncology unless deficiency is documented [5,11,18]. Accordingly, decisions about B12 use should be based on baseline status and clear clinical indication rather than empirical supplementation.

Although we did not experimentally examine downstream signaling pathways in our models, prior preclinical and translational studies indicate that G-CSF can activate pro-survival signaling pathways (STAT3 and PI3K/AKT), induce the epithelial–mesenchymal transition (EMT), and expand myeloid-derived suppressor cells, thereby fostering an immunosuppressive tumor microenvironment [19]. Clinically, findings are context-dependent: in extensive-stage small-cell lung cancer, a recent retrospective analysis suggested that concomitant G-CSF could attenuate chemo-immunotherapy efficacy, whereas another cohort reported no significant impact on outcomes. These contrasting results underscore the need for biomarker-informed, disease-specific evaluation [20,21].

Clinical observations have associated certain supplementation practices—including B vitamin supplementation—with poorer chemotherapy outcomes in some cohorts. Emerging preclinical and translational evidence suggests that G-CSF can promote tumor progression and treatment resistance in defined contexts [14,22,23].

Several preclinical studies have investigated the complex relationship between B12 exposure and cancer biology, though the evidence remains inconclusive. Large observational and clinical datasets have generally not shown a consistent association between elevated circulating B12 and cancer progression across tumor types, with liver cancer representing a notable exception—likely due to the liver’s central role in B12 metabolism and storage [23,24,25]. To our knowledge, this is the first study to examine the impact of B12 on TNBC cells specifically. We demonstrated that B12, at concentrations ranging from nanomolar to micromolar levels, significantly enhanced proliferation and viability in MDA-MB-231 cells. At 500 µM (suprapharmacological) B12, cell viability exceeded 110% at 24 h, indicating a modest but reproducible mitogenic effect. Importantly, co-treatment with B12 attenuated cisplatin-induced cytotoxicity in TNBC cells, as evidenced by a ~30–40% reduction in LDH release compared to cisplatin monotherapy. These protective effects were further supported by decreased activation of caspase-3, -8, and -9 and preserved mitochondrial membrane potential, suggesting that B12 interferes with both intrinsic and extrinsic apoptotic signaling pathways.

Prior reports indicate that depriving cells of cobalamin disrupts DNA synthesis and curtails proliferation, nominating B12 pathways as potential therapeutic targets in select cancers [2,3]. Rzepka et al. further showed that pharmacologic B12 depletion—achieved with a potent cobalamin antagonist—suppresses glioblastoma growth in vitro and in vivo by provoking G2/M arrest [26]. In contrast to depletion models, we perturbed the same axis in the opposite direction, escalating exogenous B12. This maneuver increased proliferation/viability and blunted cisplatin cytotoxicity—with preserved mitochondrial membrane potential and attenuated activation of caspases-3, -8, -9—in both MCF-7 and MDA-MB-231 cells. While B12 alone was not cytotoxic, small gains in viability at specific doses, together with mitigation of the effects of cisplatin, support a pro-survival role and align with observations that elevated serum B12 in cancer can reflect tumor-associated transcobalamin overexpression that sustains malignant cell programs [5]. Viewed alongside the depletion data, our findings suggest that extremes in cobalamin balance exert bidirectional control over tumor behavior (deficiency → growth restraint; excess → proliferation and chemoresistance), reinforcing that cobalamin’s effects are context-contingent and shaped by tissue type, molecular subtype, and treatment setting.

TNBC, especially its basal-like subtypes, displays high genomic instability due to deficiencies in DNA repair mechanisms, notably homologous recombination. This inherent vulnerability is thought to underlie the increased sensitivity of TNBC cells to DNA-damaging agents, including platinum-based chemotherapies [27,28,29]. Although prospective studies have demonstrated the clinical efficacy of cisplatin in both early-stage and metastatic TNBC [30], its role in routine clinical management remains debated due to inconsistent outcomes and the retrospective nature of much of the supporting evidence [31]. Multiple ongoing clinical trials are currently investigating the utility of cisplatin as a monotherapy or in combination regimens for TNBC. Consistent with previous preclinical reports [32,33], our study confirmed the cytotoxic efficacy of cisplatin in TNBC cells in vitro. For instance, 40 µM cisplatin reduced cell viability in MDA-MB-231 cells by more than 50% after 72 h. However, co-administration of B12 significantly mitigated this effect, restoring viability to ~75% and reducing LDH-released cytotoxicity by nearly half. These results indicate that B12 exposure alters the cellular stress response to DNA damage, potentially through epigenetic adaptation or mitochondrial stabilization [34,35]. Mechanistically, B12 sustains one-carbon metabolism via methionine synthase, maintaining cellular S-adenosylmethionine pools for DNMT (DNA methyltransferase)-mediated DNA methylation [34,36,37]; upregulation of the cobalamin transport machinery (e.g., *CD320/TCblR* and the transcobalamin axis) in malignant cells may further amplify these effects [3,4,10]. In this context, a key implication of our findings is the identification of B12-associated cisplatin resistance in TNBC cells. This phenomenon warrants further pharmacogenomic and epigenetic investigation to delineate subgroups that might benefit from cobalamin-modulating strategies (including anti-cobalamin approaches) [2,26] and contraindicates empirical B12 supplementation during platinum-based chemotherapy in the absence of baseline deficiency.

A major additional finding is that G-CSF promotes proliferation and reduces cisplatin sensitivity in TNBC cells. Treatment of MDA-MB-231 cells with 50 ng/mL G-CSF for 48 h increased viability by >130% compared to untreated controls. The effect was time- and dose-dependent, peaking at 48–72 h. In cisplatin-treated groups, G-CSF reduced LDH release by ~40% and suppressed caspase activation while restoring mitochondrial membrane potential. These data indicate that G-CSF interferes with both intrinsic and extrinsic apoptosis pathways. Interestingly, co-administration of B12 and G-CSF suppressed cisplatin-induced upregulation of cytotoxic effectors such as perforin and granzymes A/B, raising the possibility that these agents may interfere with immunogenic cell death, which warrants validation in immunocompetent models [38]. Because perforin and granzymes are classically lymphocyte-derived, their detection in tumor cell lysates under monoculture conditions should be interpreted cautiously and validated with orthogonal approaches (e.g., transcript assays and antibody-specificity controls).

Although G-CSF is widely used to manage chemotherapy-induced neutropenia, growing evidence suggests that it has pro-tumorigenic effects. Tumor-derived or elevated G-CSF has been described in several solid tumors and has been associated with aggressive clinical behavior [14,19,39]. G-CSF may also promote tumor progression via *CSF3R*-coupled JAK/STAT and PI3K/AKT signaling pathways, the epithelial–mesenchymal transition (EMT), and expansion of myeloid-derived suppressor cells (MDSCs), ultimately creating an immunosuppressive microenvironment [40,41,42]. In a bladder cancer model, Hori et al. showed that G-CSF promotes tumor progression by enhancing angiogenesis, M2 macrophage polarization, and the EMT [43]. Additionally, case reports in breast cancer have demonstrated that elevated serum G-CSF levels correlate with tumor burden and regress following resection of G-CSF-secreting tumors [44,45]. To our knowledge, few studies have investigated exogenous, pharmacologic G-CSF exposure in TNBC; our data further show that G-CSF enhances proliferation and attenuates cisplatin efficacy in vitro. These observations warrant careful consideration of the use of G-CSF in aggressive breast cancer subtypes, particularly when apoptosis-reliant regimens such as cisplatin are employed. Although our pathway references are literature-based and were not experimentally interrogated in this study, the preservation of mitochondrial integrity and the suppression of caspase activation we observed are consistent with the activation of pro-survival signaling downstream of *CSF3R*. Accordingly, tumors with high *CSF3R* expression (rather than assuming activated mutations) may be particularly susceptible to G-CSF-driven proliferative and cytoprotective effects; future translational studies should evaluate *CSF3R* status as a candidate biomarker of G-CSF-associated chemoresistance.

In vivo, G-CSF can expand myeloid-derived suppressor cells (MDSCs) and skew macrophage polarization toward pro-tumorigenic phenotypes, creating an immunosuppressive microenvironment [19,41]. Although our monoculture system lacked an immune component, the preservation of mitochondrial integrity and inhibition of caspases suggest that G-CSF can directly confer resistance to DNA-damaging agents [19,39,40,46]. The findings of our study indicate that exogenous administration of G-CSF promotes cell proliferation and induces resistance to cisplatin in breast cancer cell lines, including both MCF-7 and MDA-MB-231 subtypes. In both models, treatment with pharmacologic concentrations of G-CSF (10–50 ng/mL) resulted in a significant time- and dose-dependent increase in cell viability, with peak effects observed at 48–72 h. Notably, co-treatment with G-CSF markedly reduced cisplatin-induced cytotoxicity, as evidenced by reduced LDH release, diminished caspase activation, and preservation of mitochondrial membrane potential. These findings raise important translational concerns regarding the use of G-CSF, which is routinely employed for neutropenia prophylaxis in clinical oncology. These preclinical data do not alter guideline-directed indications for G-CSF; rather, they support a biomarker-informed, context-specific risk–benefit appraisal when integrating supportive agents with DNA-damaging chemotherapy.

Several limitations of our study must be acknowledged. First, our experiments were performed in vitro without immune cells, precluding the assessment of systemic immunologic effects such as leukocytosis or potential changes in circulating B12. Therefore, future studies in immunocompetent in vivo models are warranted to validate these findings and to better delineate the interactions of B12 and G-CSF with chemotherapy within the tumor immune microenvironment. Second, a subset of the concentrations explored—specifically 500 µM B12 and 50 µg/mL G-CSF—exceed clinically observed plasma levels; these exposures should be regarded as suprapharmacological and exploratory, and their translational relevance should be interpreted with caution. Third, the *BRCA1/2* mutation status of the cell lines, which may influence platinum sensitivity, was not determined. Fourth, upstream receptors (*CSF3R* and *TCN2*) and downstream pathways (STAT3 and PI3K/AKT) were not interrogated. These mechanisms are highly likely to underlie the observed cytoprotective effects and should be addressed in future mechanistic studies, which will require dedicated experimental designs with more advanced resources than were available during this study. Our analyses focused on apoptotic regulators because the study was designed to investigate cisplatin-induced cell death. Future studies will extend this approach by incorporating receptor- and signaling-level analyses, including protein-level validation (e.g., Western blot of perforin, granzymes, and upstream regulators), which will be planned and budgeted for. Another limitation of the present work is that representative images of morphology were not systematically captured at different treatment doses; future studies should incorporate imaging-based assessments to better contextualize phenotypic alterations alongside viability data.

Nevertheless, our study highlights that agents typically considered benign or beneficial in supportive care—such as G-CSF and B12—may exert pro-survival effects on tumor cells under certain molecular conditions. Importantly, these effects were observed across distinct molecular subtypes of breast cancer, suggesting a broader relevance. These findings underscore the need for caution when administering supportive agents during cytotoxic therapy and call for further preclinical and clinical studies to determine the safety and implications of their use in oncology.

## 4. Materials and Methods

### 4.1. Chemicals and Reagents

Vitamin B12 (Sigma Aldrich, St. Louis, MO, USA; Cas No: 68-19-9), G-CSF (G-CSF; Sigma Aldrich; G0407), cisplatin (Merck, St. Louis, MO, USA; Cas No: 15663-27-1), 3-(4,5-dimethylthiazol-2-yl)-2,5-diphenyltetrazolium bromide (MTT; Merck; Cas No: 298-93-1), and dimethyl sulfoxide (DMSO; Sigma Aldrich; Cas No: 67-68-5) were obtained from Sigma-Aldrich and Merck (St. Louis, MO, USA). B12 was applied at concentrations ranging from 0.0005 to 500 µM [26], G-CSF (recombinant human) was applied at 0.001 to 50 µg/mL [46], and cisplatin was applied at 5 to 500 µM [32]. The concentration ranges were selected to include both clinically relevant levels (B12: 0.0005–1 µM; G-CSF: 1–50 ng/mL) and higher exploratory exposures (up to 500 µM for B12 and 50 µg/mL for G-CSF, equivalent to 1–50,000 ng/mL). The clinical Cmax after subcutaneous G-CSF administration is reported to be ~1–49 ng/mL [47]. This design was intended to capture the full dose–response spectrum.

The breast cancer cell lines MCF-7 (*p53*+ and ER+) and MDA-MB-231 (*p53* mutant and ER-) were supplied by American Type Cell Culture Collection (ATCC, Manassas, VA, USA). Dulbecco’s Modified Eagle Medium (DMEM; Ref: 01-052-1A), fetal bovine serum (FBS; Ref: 04-007-1A), L-glutamine (Ref: 03-020-1B), penicillin–streptomycin (Ref: 03-031-1B), and trypsin (Ref: 03-015-1B) were obtained from Biological Industries. A Cytotoxicity Detection Kit Plus was provided by Roche Applied Science (Basel, Switzerland) (Ref: 04744926001), and the activities of caspase-3, caspase-8, and caspase-9 were evaluated using a fluorometric multiplex activity assay kit (Abcam, Cambridge, UK, ab219915). A Mitochondrial Membrane Potential Assay Kit II was purchased from Cell Signaling Technology (Danvers, MA, USA) (CST, 13296), and a BCA Protein Assay Kit was obtained from Thermo Scientific (Waltham, MA, USA) (Ref: 23227). All Western blot antibodies were obtained from CST. Reagents for protein gel electrophoresis were purchased from Bio-Rad (Hercules, CA, USA), and ECL Western blot Substrate was obtained from Millipore (Burlington, MA, USA) (Cat. No: WBLUR0500). Two breast cancer models were selected to capture divergent biology: MCF-7 (ER/PR-positive; luminal-like; wild-type *p53*) and MDA-MB-231 (triple-negative; mesenchymal-like; mutant *p53*). This pairing represents clinically relevant subtypes with distinct apoptotic and stress-response pathways, enabling evaluation of whether B12 and G-CSF modulate cisplatin responses across contrasting molecular contexts.

### 4.2. Cell Culture

The human breast cancer cell lines MDA-MB-231 and MCF-7 were cultured in Dulbecco’s Modified Eagle Medium (DMEM) supplemented with 10% fetal bovine serum (FBS) and 1% penicillin–streptomycin and maintained in a humidified incubator at 37 °C with 5% CO_2_. The cells were passaged every 2–3 days at 70–80% confluency and seeded 24 h before treatment in either 96-well plates (2 × 10^4^ cells/100 μL per well) or 6-well plates (5 × 10^6^ cells/2 mL per well).

### 4.3. Cell Viability Assessment

The MTT test was used to assess changes in cell viability in the groups following 24, 48, and 72 h of both single and combined drug treatments. The assay was performed by replacing the cell media with complete media containing 0.5 mg/mL MTT and incubating the cells at 37 °C in a 5% CO_2_ humidified atmosphere for 4h. Following incubation, the MTT media were aspirated, and the resulting formazan crystals were solubilized by adding 100 μL of DMSO to each well. Endpoint absorbance at 570 nm was measured using a Spectramax M3 microplate reader (Molecular Devices, San Jose, CA, USA). Each condition was tested in six replicates in three independent experiments [48].

### 4.4. Cytotoxicity Assay

Cytotoxicity was assessed using the lactate dehydrogenase (LDH) assay at 24, 48, and 72 h post drug treatment. The manufacturer’s protocol was followed, and in the final step, the optical density (OD) was measured at 490 nm with a Spectramax M3 microplate reader (Molecular Devices, CA, USA). Each experiment was conducted in three separate trials with six replicates [48].

### 4.5. Caspase Activity Assays

The activity of caspase-3, -8, and -9 was assessed using the Multi-Activity Assay Kit (Abcam, ab219915, UK), following the manufacturer’s instructions. The cells were added to 100 μL/well of caspase assay solution after a 24-h combination drug treatment and incubated for 45 min at room temperature. Fluorescence intensities were measured at Ex/Em wavelengths for each caspase: Ex/Em = 535/620 nm for Caspase-3, Ex/Em = 490/525 nm for Caspase-8, and Ex/Em = 370/450 nm for Caspase-9.

### 4.6. Mitochondrial Membrane Potential Measurement

Mitochondrial membrane potential was assessed using the TMRE Assay Kit (CST, 13296, Danvers, MA, USA) according to the manufacturer’s instructions. After the combined drug interaction, the cell lines were incubated with 200 nM of TMRE for 20 min and maintained at 37 °C in an incubator before analysis. The cells were washed three times with warmed 1X PBS and then incubated in pre-warmed 1X PBS. In the final step of the assay, the reader settings included excitation at about 550 nm and emission at 580 nm.

### 4.7. Western Blot Analysis

Proteins were extracted from the cells with RIPA lysis buffer on ice and quantified using a bicinchoninic acid (BCA) assay kit (Thermo Scientific). Protein lysates were separated with 10–12% SDS-PAGE (sodium dodecyl sulfate–polyacrylamide gel electrophoresis)–polyacrylamide gels and then transferred onto polyvinylidene fluoride (PVDF, Millipore) membranes. After incubation overnight with primary antibodies (granzyme A (1:1000); granzyme B (1:1000), perforin (1:1000), and β-actin (1:1000)), the membranes were treated for 90 min with secondary antibodies conjugated to horseradish peroxidase (HRP). Immunoreacted bands were detected using a Chemi^Doc^ Kodak Gel Logic 2200 Pro. (Carestream Health, Rochester, NY, USA) through the ECL detection system (Millipore). The optical density of the bands was quantified using ImageJ software (version 1.53t) (National Institutes of Health, Bethesda, MD, USA), and β-actin was used as a loading control [48].

### 4.8. Statistical Analysis

Statistical analyses were performed using SigmaStat v3.5 software (Systat Software, Inc., San Jose, CA, USA). All MTT data were expressed as the mean values ± standard deviation (SD) and analyzed via Student’s *t*-test. The results of the viability and LDH release assays (with Bonferroni post hoc test) were analyzed using an analysis of variance test. Combination index curves were calculated and visualized using Compusyn (BioSoft, El Cajon, CA, USA) [49]. All data are presented as the mean ± standard deviation, and values of *p* < 0.05 were considered statistically significant.

## 5. Conclusions

Our study demonstrates that pharmacologic exposure to B12 and G-CSF can enhance breast cancer cell proliferation and attenuate cisplatin-induced cytotoxicity in vitro. These effects were observed in both MCF-7 and MDA-MB-231 breast cancer models and were associated with reduced apoptotic signaling and preservation of mitochondrial integrity. These findings raise important concerns regarding the empirical use of supportive care agents during chemotherapy.

Specifically, our data suggest that a subset of patients may be vulnerable to B12-associated chemoresistance, highlighting the need to maintain serum B12 levels within physiological ranges and to avoid unnecessary supplementation, particularly in cases of triple-negative breast cancer. These results also support the exploration of anti-cobalamin strategies as a potential therapeutic approach in biologically defined subgroups. Furthermore, our preclinical observation that exogenous G-CSF promotes tumor cell survival and attenuates cisplatin efficacy supports a biomarker-informed, guideline-directed risk–benefit appraisal within established indications for neutropenia prophylaxis. The clinical implications remain incompletely defined, and particular caution may be warranted in patients with G-CSF-producing tumors or *CSF3R*-positive disease, pending biomarker-driven prospective studies. If confirmed by future in vivo and clinical studies, our findings may guide more personalized supportive care strategies and prevent inadvertent tumor protection. Overall, our study emphasizes the importance of integrating tumor biology and pharmacologic context into the design of both therapeutic and supportive oncology regimens.

## Figures and Tables

**Figure 1 ijms-26-09086-f001:**
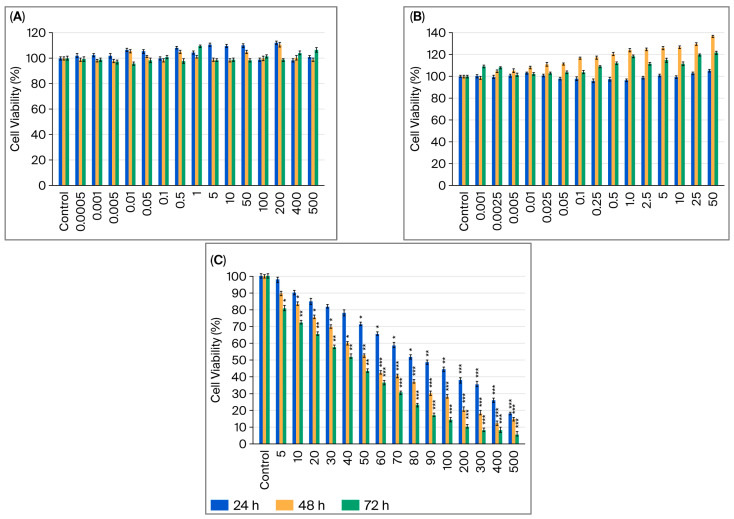
MCF-7 cell viability after 24, 48, and 72 h of exposure to B12 (0.0005–500 µM) (**A**); G-CSF (0.001–50 µg/mL) (**B**); cisplatin (5–500 µM) (**C**). Data are expressed as means ± SDs (standard deviation) (*n* = 6). * *p* < 0.05, ** *p* < 0.01, and *** *p* < 0.001 compared with control group.

**Figure 2 ijms-26-09086-f002:**
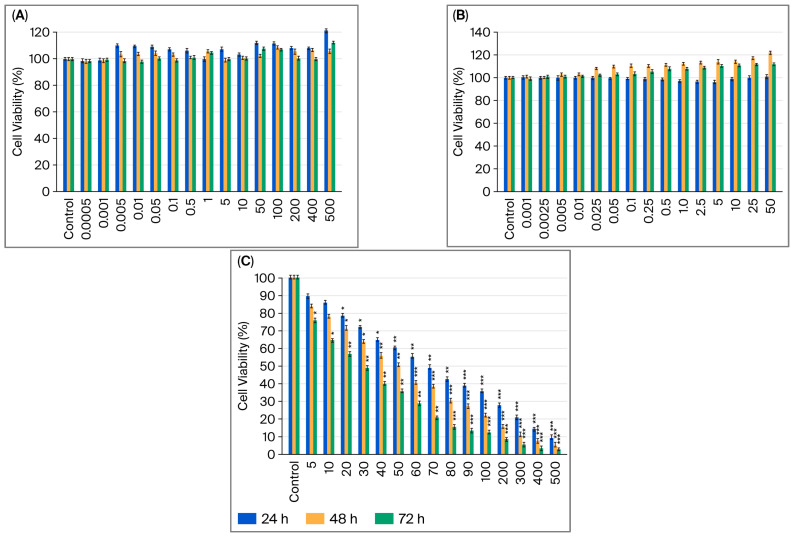
MDA-MB-231 cell viability after 24, 48, and 72 h of exposure to B12 (0.0005–500 µM) (**A**); G-CSF (0.001–50 µg/mL) (**B**); cisplatin (5–500 µM) (**C**). Data are expressed as means ± SDs (*n* = 6). * *p* < 0.05, ** *p* < 0.01, and *** *p* < 0.001 compared with control group.

**Figure 3 ijms-26-09086-f003:**
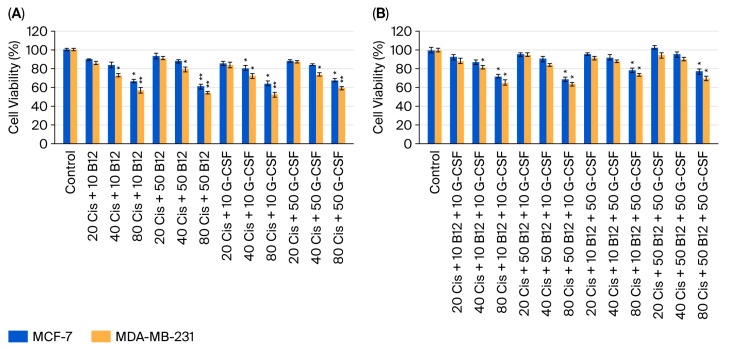
Effect of dual (**A**) and triple (**B**) drug combinations on cell viability in MCF-7 and MDA-MB-231 breast cancer cell lines, determined by MTT assay. Data are expressed as means ± SDs (*n* = 6). * *p* < 0.05; ** *p* < 0.01 vs. control. Cis, cisplatin (20, 40, 80 µM); B12 (10 or 50 nM); G-CSF (10 or 50 ng/mL).

**Figure 4 ijms-26-09086-f004:**
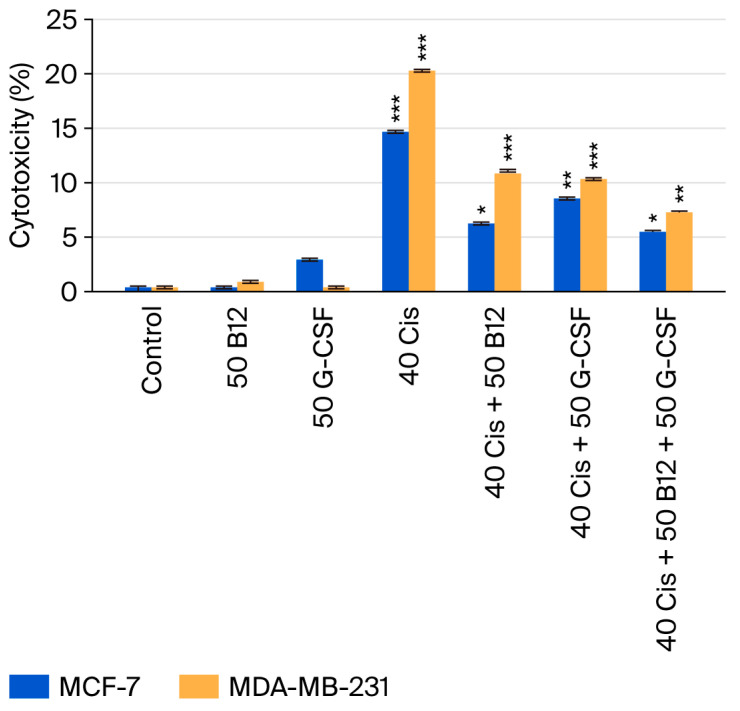
Effect of cisplatin (Cis) alone or in combination with B12 and/or G-CSF on cytotoxicity in MCF-7 and MDA-MB-231 breast cancer cell lines, determined by LDH release assay. Data are expressed as means ± SDs (*n* = 3). * *p* < 0.05; ** *p* < 0.01; *** *p* < 0.001 vs. control. Cis, cisplatin (40 µM); B12 (50 nM); G-CSF (50 ng/mL); LDH, lactate dehydrogenase.

**Figure 5 ijms-26-09086-f005:**
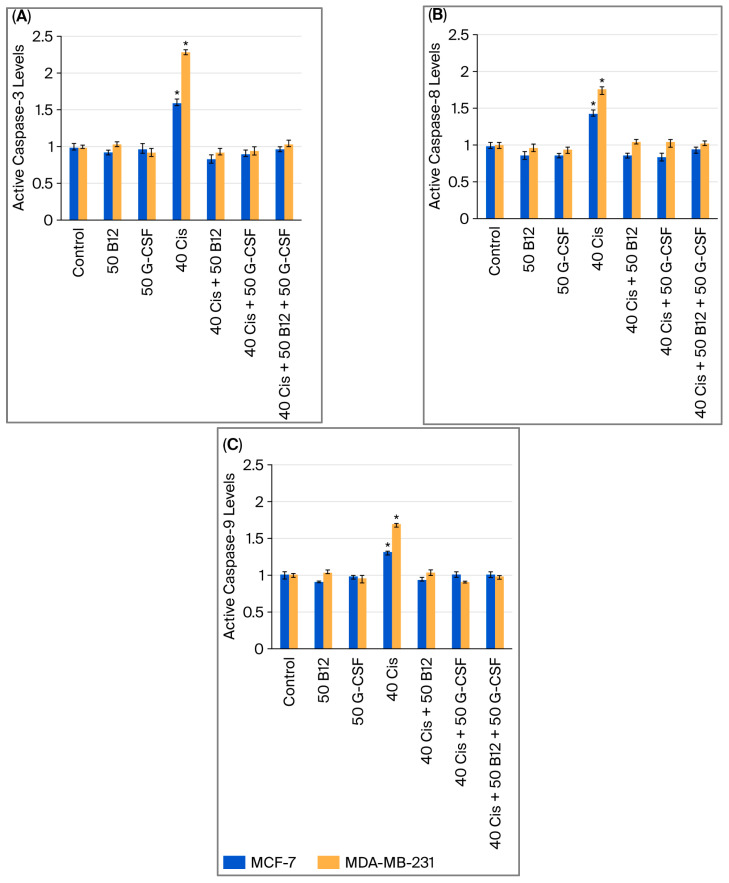
Activation of caspase-3 (**A**), caspase-8 (**B**), and caspase-9 (**C**) in MCF-7 and MDA-MB-231 cells following 24 h of treatment with cisplatin (40 µM) alone or in combination with B12 (50 nM) and/or G-CSF (50 ng/mL), quantified using fluorometric multiplex caspase activity assay. Data are expressed as means ± SDs (*n* = 3). * *p* < 0.05 vs. control.

**Figure 6 ijms-26-09086-f006:**
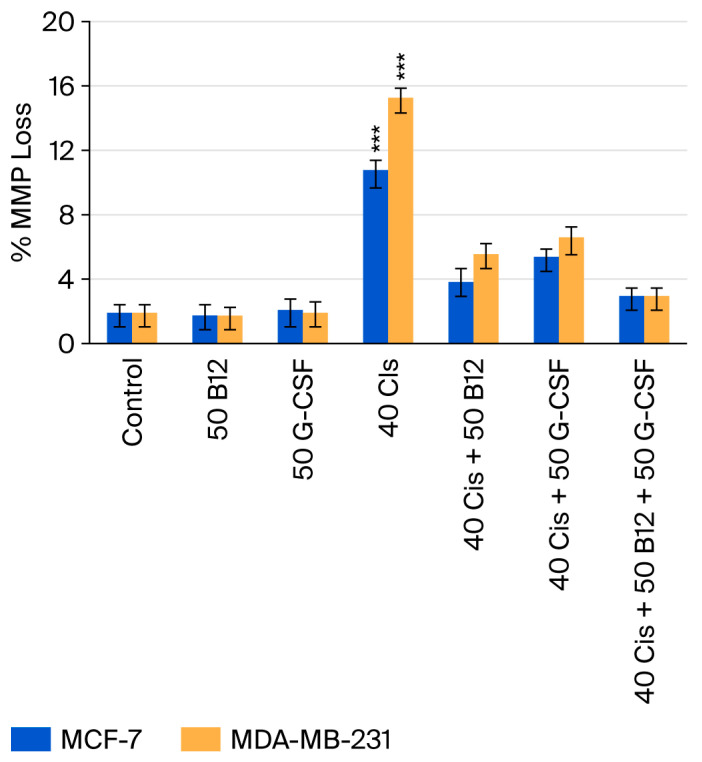
Loss of mitochondrial membrane potential (MMP) in MCF-7 and MDA-MB-231 cells following treatment with cisplatin (40 µM) alone or in combination with B12 (50 nM) and/or G-CSF (50 ng/mL), assessed using the TMRE assay. Data are expressed as means ± SDs (*n* = 3). *** *p* < 0.001 vs. control.

**Figure 7 ijms-26-09086-f007:**
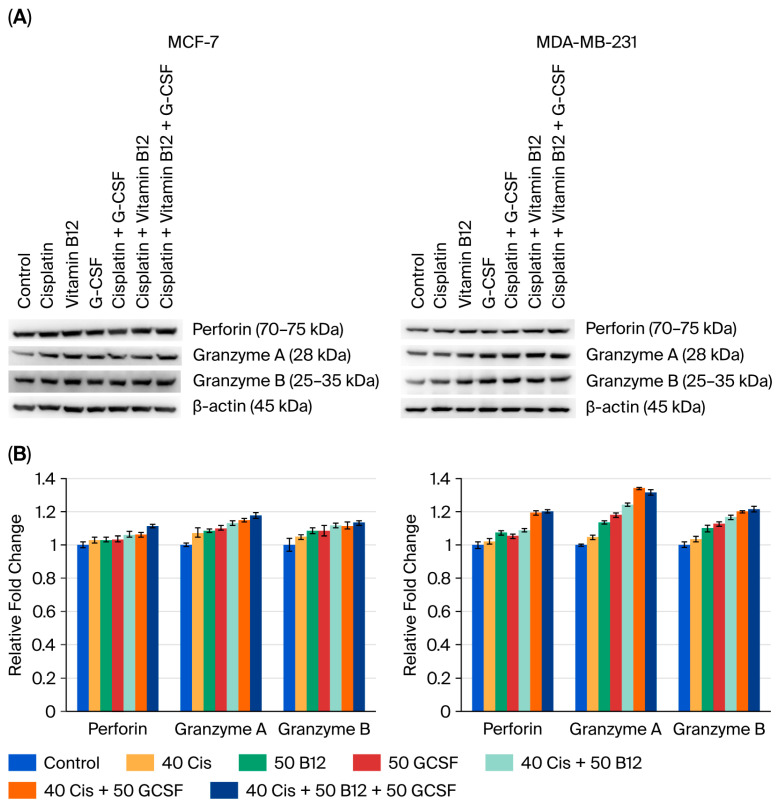
Effects of B12 and G-CSF on cisplatin-induced perforin and granzyme expression in MCF-7 and MDA-MB-231 cells. Western blot images of perforin, granzyme A, granzyme B, and β-actin in MCF-7 and MDA-MB-231 cells treated with cisplatin, B12, G-CSF, or their combinations (**A**). Relative protein expression of perforin, granzyme A, and granzyme B in both cell lines (**B**). Data are expressed as means ± SDs (*n* = 3).

**Figure 8 ijms-26-09086-f008:**
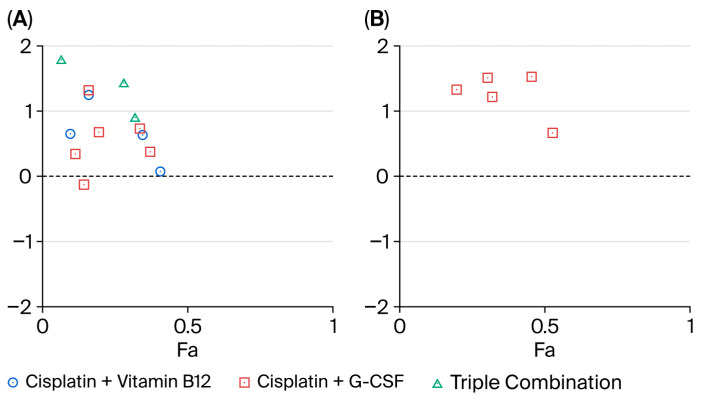
Combination Index (CI) versus fraction affected (Fa) plots for cisplatin-based combinations in breast-cancer cells. (**A**), MCF-7; (**B**), MDA-MB-231. Symbols denote experimental CI values at the observed Fa levels: blue circles, B12 + cisplatin; red squares, G-CSF + cisplatin; green triangles, B12 + G-CSF + cisplatin. Cell viability was measured by MTT, normalized to vehicle, and converted to effect level (Fa = 1 − viability/100). CI values were computed in CompuSyn using Chou–Talalay method under a non-constant ratio design (dose order: B12–G-CSF–cisplatin). Horizontal reference marks CI = 1 (additivity); CI < 1 indicates synergy; CI > 1 antagonism. Points are shown without trendlines to emphasize observed CI values across tested Fa range (≈0–0.5).

## Data Availability

The original contributions presented in this study are included in the article. Further inquiries can be directed to the corresponding author.

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
