# Peer review of "The Responsiveness of Breast Cancer Cells to Varied Levels of Vitamin B12, Cisplatin, and G-CSF"

_ijms, 2025, doi:10.3390/ijms26189086_

Round 1
Reviewer 1 Report
Comments and Suggestions for Authors
MMy overall assessment is Major Revision. The manuscript addresses a novel and clinically relevant question, presents a comprehensive in vitro design, and offers a timely warning about the possible tumor-promoting effects of routine supportive-care drugs. Nevertheless, methodological details, mechanistic depth, and translational relevance need substantial improvement before the work can be accepted for publication.
- The highest concentrations used (500 µM B12; 50 µg/mL G-CSF) far exceed clinically achievable Cmax values (B12 ≈ 1 nM–1 µM; G-CSF s.c. Cmax ≈ 20 ng/mL).
- The study is largely descriptive; upstream receptors (CSF3R, TCN2) and downstream signaling (STAT3, PI3K/AKT) are not interrogated.
- The Discussion under-represents relevant clinical data.
- The Abstract and Results contain grammatical and spelling inconsistencies (e.g., “GCSF” vs “G-CSF”).
- Results obtained in vitro require confirmation in an immunocompetent setting.
- I have concerns regarding the Western blot results. Please provide the original, uncropped images that include the molecular-weight markers.
Author Response
Comments from reviewers 1
Comment 1. The highest concentrations used (500 µM B12; 50 µg/mL G-CSF) far exceed clinically achievable Cmax values (B12 ≈ 1 nM–1 µM; G-CSF s.c. Cmax ≈ 20 ng/mL).
Response:
Thank you for your valuable contributions and insights regarding our study. We acknowledge that the upper concentrations tested (500 µM vitamin B12 and 50 µg/mL G-CSF) exceed the clinically reported Cmax values. We emphasize that our main combination experiments (B12 at 0.01–0.05 µM(10 or 50 nM) and G-CSF at 10–50 ng/mL) fully overlapped with clinically achievable plasma concentrations, and the protective effects were already significant at these ranges In the context of academic research in our country, we face significant challenges due to limited budgets and the considerable depreciation of our national currency against other currencies. These constraints necessitate a strategic approach to maximize the scientific output of our studies. Accordingly, the current study was designed to generate comprehensive preliminary data to inform future research efforts. To this end, we employed a broad dose range to thoroughly characterize the dose–response profile, enabling the identification of potential threshold that may not be evident within narrower concentration ranges. Cmax concentrations were included within the range we studied, and these doses were selected for the combination studies. However, we sincerely appreciate your valuable comments, which helped us identify an inadvertent error. The use of 10 nM and 50 nM concentrations of vitamin B12 was mistakenly reported as micromolar. This error has been corrected throughout the manuscript. Due to the reasons outlined, an extensive dose-response study was conducted for G-CSF. Based on the literature and the prescribing information from the FDA approval, the study proceeded with concentrations of 10 ng/mL and 50 ng/mL [1].
At these clinically relevant ranges, we already observed significant proliferative and cytoprotective effects. In addition, the highest dose of G-CSF we used (50 ng/mL) in the clinically relevant range is consistent with previously published in vitro studies, including Ref. [16], which applied G-CSF at 50 ng/mL. This dose selection was also supported by Ref. [16] and cited in our Methods section.
We identified typographical unit errors in the Figure 3 legend. The correct concentrations used were vitamin B12: 10 or 50 nM (0.01 or 0.05 µM) and G-CSF (10 or 50 ng/mL), consistent with the Methods and Results sections. This has now been corrected in the revised Figure 3 legend.
To address this point, we have:
- Clarified the rationale for the selected concentration ranges in the Methods section (page 2, lines 79-83), explicitly noting the equivalent units and the corresponding clinical reference values and highlighted in the revised manuscript
- Added a statement in the Discussion acknowledging that the highest concentrations represent a limitation of the study and should be interpreted with caution regarding clinical translation (page 16-17, lines 486-489) and highlighted in the revised manuscript
Accordingly, the following sentences were added to the manuscript:
- 1. Concentration ranges were selected to include both clinically relevant levels (B12: 0.0005–1 µM; G-CSF: 1–50 ng/mL) and higher exploratory exposures (up to 500 µM for B12 and 50 µg/mL for G-CSF, equivalent to 1–50,000 ng/mL). The clinical Cmax after subcutaneous G-CSF administration is reported as ~1–49 ng/mL [1]. This design was intended to capture the full dose–response spectrum.
- We acknowledge that the upper concentrations tested (500 µM B12 and 50 µg/mL G-CSF) exceed clinically observed plasma levels. These doses should be considered supra-pharmacological and exploratory, and their translational relevance is limited.
We think the Methods and Discussion sections have become much better thanks to your suggestions.
Comment 2. The study is largely descriptive; upstream receptors (CSF3R, TCN2) and downstream signaling (STAT3, PI3K/AKT) are not interrogated.
Response:
We sincerely appreciate this valuable comment. We agree with the reviewer’s observation that our study did not investigate upstream receptors, such as CSF3R and TCN2, or downstream signaling pathways, including STAT3 and PI3K/AKT. In our manuscript, references to JAK/STAT and PI3K/AKT were based solely on prior literature to contextualize our findings. We did not experimentally interrogate these pathways in our models. We have now clarified this point in the Discussion to avoid any misinterpretation and to highlight that such mechanistic analyses will be the focus of our future studies.
The primary objective of this study was to provide preliminary in vitro evidence exploring the potential effects of vitamin B12 and G-CSF in the context of cisplatin treatment, while highlighting their possible translational implications for future research. We acknowledge that mechanistic analyses such as receptor expression and pathway activation (e.g., phosphorylation of STAT3 or AKT) would significantly enrich the findings. However, the project was conducted under considerable economic and logistical constraints, and therefore we prioritized a comprehensive set of functional assays (viability, apoptosis, mitochondrial function, LDH release) to establish a foundation for future research. Because our study was specifically designed around cisplatin treatment, we focused on assessing the expression of proteins involved in the initial apoptotic pathways to understand how vitamin B12 and G-CSF may modulate cisplatin-induced cell death. In future projects, we plan to integrate receptor- and pathway-level analyses (e.g., CSF3R/STAT3, TCN2/PI3K/AKT) into our experimental design, provided that appropriate funding and resources are available.
More detailed receptor- and pathway-level investigations were unfortunately beyond the scope of the present study. Nevertheless, in this study, which serves as a preliminary foundation for our subsequent research, we endeavored to analyze, with limited resources, the effects of combined drug treatments on the levels of key proteins in the intrinsic, extrinsic and perforin/granzyme pathways using Western blot analysis.
Additionally, we recognize that CSF3R/STAT3 and TCN2/PI3K/AKT axes are very likely to play central roles in the observed effects. We have now added a statement in the Discussion to highlight this limitation and to emphasize that elucidating these pathways will be the focus of our future mechanistic and translational studies, provided that sufficient funding is available.
Accordingly, the following sentence was added to the Discussion (page 17, lines 491-498) and highlighted in the revised manuscript:
Fourth, upstream receptors (CSF3R and TCN2) and downstream pathways (STAT3 and PI3K/AKT) were not interrogated. These mechanisms are highly likely to underlie the observed cytoprotective effects and should be addressed in future mechanistic studies, which will require dedicated experimental designs with more advanced resources than were available during this study. Our analyses focused on apoptotic regulators because the study was designed to investigate cisplatin-induced cell death. Future studies will extend this approach by incorporating receptor- and signaling-level analyses, which will be planned and budgeted for.
Comment 3. The Discussion under-represents relevant clinical data.
Response:
We thank the reviewer for this valuable comment. In the revised manuscript, we have expanded the Discussion to incorporate additional clinical studies and reviews addressing the association between supraphysiological or persistently elevated vitamin B12 levels and cancer incidence or mortality, as well as preclinical and clinical data suggesting potential tumor-promoting effects of G-CSF. We also highlighted the context-dependent nature of these findings, citing both studies that reported negative effects on chemo-immunotherapy efficacy and those that did not observe significant impact on outcomes. Furthermore, the entire Discussion section was carefully restructured to improve clarity and flow, with some sentences simplified to enhance readability
Accordingly, the following paragraph was added to the Discussion (page 14, lines 347-367) and highlighted in the revised manuscript).
From a clinical perspective, multiple population-based and integrative reviews report that supraphysiologic or persistently elevated plasma vitamin B12 is associated with a higher short-term incidence of newly diagnosed cancer and with increased mortality, although concerns regarding reverse causality and confounding (e.g., liver disease, occult malignancy) remain; these data argue against unnecessary supplementation in oncology unless deficiency is documented[5, 24, 25]. Accordingly, decisions about B12 use should be based on baseline status and clear clinical indication rather than empirical supplementation. Although we did not experimentally examine downstream signaling pathways in our models, prior preclinical and translational studies indicate that G-CSF can activate pro-survival signaling pathways (STAT3 and PI3K/AKT), induce the epithelial–mesenchymal transition (EMT), and expand myeloid-derived suppressor cells, thereby fostering an immunosuppressive tumor microenvironment [26]. Clinically, findings are context-dependent: in extensive-stage small-cell lung cancer, a recent retrospective analysis suggested that concomitant G-CSF could attenuate chemo-immunotherapy efficacy, whereas another cohort reported no significant impact on outcomes. These contrasting results underscore the need for biomarker-informed, disease-specific evaluation [27, 28].
Comment 4. The Abstract and Results contain grammatical and spelling inconsistencies (e.g., “GCSF” vs “G-CSF”).
Response:
We sincerely thank the reviewer for this valuable observation. We carefully re-checked the abstract, results, and the entire manuscript for typographical, grammatical, and spelling inconsistencies. Specifically, we corrected all instances of “GCSF” to the standardized and internationally accepted form “G-CSF.” In addition, minor grammatical errors and inconsistent abbreviations were revised throughout the text to ensure clarity, accuracy, and uniformity.
Comment 5. Results obtained in vitro require confirmation in an immunocompetent setting.
Response:
We thank the reviewer for this important point and fully agree that our in vitro findings require validation in immunocompetent models. As already noted in the Discussion, we explicitly acknowledged this limitation:
“Several limitations of our study must be acknowledged. First, our experiments were performed in vitro without immune cells, precluding assessment of systemic immunologic effects such as leukocytosis or potential changes in circulating B12.”
To further clarify and strengthen this point, we have now added the following sentence in the Discussion (page 16, lines 483-486, highlighted in the revised manuscript) and highlighted in the revised manuscript
Therefore, future studies in immunocompetent in vivo models are warranted to validate these findings and to better delineate the interactions of vitamin B12 and G-CSF with chemotherapy within the tumor immune microenvironment.
We believe this addition provides a clear acknowledgment of the limitation and outlines the necessary next steps for translational validation. This will be a priority in future research, ideally through larger-scale, adequately powered, and well-funded studies.
Comment 6. I have concerns regarding the Western blot results. Please provide the original, uncropped images that include the molecular-weight markers.
Response:
We thank the reviewer for raising this important point. To ensure transparency and reproducibility, we have prepared the original, uncropped Western blot images including molecular-weight markers. All figures were submitted as per the journal's requirements during the initial article submission. To address any concerns and ensure clarity, the figures have been re-uploaded.
REFERENCES:
- Food, U.S. and A. Drug, Neupogen (filgrastim) prescribing information. 2013, FDA: Silver Spring, MD.

Reviewer 2 Report
Comments and Suggestions for Authors
The manuscript by Volkan Aslan et al. titled “The Responsiveness of Breast Cancer Cells to Varied Levels of Vitamin B12, Cisplatin, and G-CSF” is well written and presents a straightforward study with clear findings. To further strengthen the manuscript, I suggest the following points. The rationale for selecting both MCF-7 and MDA-MB-231 cell lines should be clearly explained, as their distinct biological characteristics may influence the outcomes. In Figure 1, it would be valuable to include representative images of cell morphology at the lowest and highest treatment doses to highlight any phenotypic differences, including potential changes in ploidy. For all figures, treatment conditions should be explicitly labeled within the images themselves, in addition to the figure legends, to aid clarity for readers. The methods section should include detailed information on all reagents and kits used, including catalog and lot numbers, to ensure reproducibility. Finally, the study would benefit from a western blot that identifies key signaling molecules and broader discussion of other signaling pathways potentially regulated by these treatments, particularly upstream regulators and transcription factors involved in mediating cytotoxic mechanisms such as perforin and granzymes.
Author Response
Reviewer 2
Comment 1. The rationale for selecting both MCF-7 and MDA-MB-231 cell lines should be clearly explained, as their distinct biological characteristics may influence the outcomes.
Response:
We thank the reviewer for this important suggestion. We have clarified our rationale for choosing MCF-7 (hormone receptor–positive, luminal-like, wild-type p53) and MDA-MB-231 (triple-negative, mesenchymal-like, mutant p53) cell lines. These models represent two clinically relevant and biologically divergent breast cancer subtypes with distinct stress-response and apoptotic programs, which are known to influence responsiveness to DNA-damaging agents such as cisplatin. Including both lines allowed us to assess whether vitamin B12 and G-CSF exert comparable cytoprotective effects across contrasting molecular contexts.
Importantly, our results support this rationale: the cytoprotective effects of vitamin B12 and G-CSF were more pronounced in MCF-7 cells compared to MDA-MB-231, which may reflect their distinct molecular characteristics (e.g., ER signaling and p53 status).
Accordingly, the following insertions were added to the manuscript (highlighted in the revised version):
Methods → Cell lines and culture (page 3, lines 95–100):
Two breast cancer models were selected to capture divergent biology: MCF-7 (ER/PR-positive; luminal-like; wild-type p53) and MDA-MB-231 (triple-negative; mesenchymal-like; mutant p53). This pairing represents clinically relevant subtypes with distinct apoptotic and stress-response pathways, enabling evaluation of whether vitamin B12 and G-CSF modulate cisplatin responses across contrasting molecular contexts.
Discussion (page 13, 333-338 lines ):
Using both a luminal/ER-positive (MCF-7, wild-type p53) and a triple-negative/mesenchymal-like model (MDA-MB-231, mutant p53) provided complementary contexts in which to interpret the cytoprotective effects of vitamin B12 and G-CSF against cisplatin. In line with this rationale, the protective effects were more pronounced in MCF-7 cells, consistent with their distinct molecular features.
We think the Methods and Discussion sections have become much better thanks to your suggestions.
Comment 2. In Figure 1, it would be valuable to include representative images of cell morphology at the lowest and highest treatment doses to highlight any phenotypic differences, including potential changes in ploidy.
Response:
We thank the reviewer for this valuable suggestion. Unfortunately, representative morphology images were not systematically captured during the original experiments and therefore cannot be added retrospectively. To address this important point, we plan to incorporate systematic morphological imaging in our future studies to provide additional context on phenotypic changes. We have also added a statement in the Discussion acknowledging the absence of representative morphology images as a limitation of the present study.
Accordingly, the following sentence was added to the Discussion (page 17, lines 498-501, highlighted in the revised manuscript):
Another limitation of the present work is that representative images of morphology were not systematically captured at different treatment doses; future studies should incorporate imaging-based assessments to better contextualize phenotypic alterations alongside viability data.
We think the Discussion sections have become much better thanks to your suggestions
Comment 3. For all figures, treatment conditions should be explicitly labeled within the images themselves, in addition to the figure legends, to aid clarity for readers.
Response:
We thank the reviewer for this practical suggestion. The figures were arranged in accordance with the reviewer’s instructions during manuscript preparation. However, because of the large amount of data, direct labeling within the figure panels would have resulted in a very crowded and unclear display. To maintain clarity, detailed explanations of the treatment conditions were therefore added below each figure.
Comment 4. The methods section should include detailed information on all reagents and kits used, including catalog and lot numbers, to ensure reproducibility
Response:
We thank the reviewer for this important suggestion. We fully agree that detailed reporting of reagents and kits is essential for reproducibility. Accordingly, we have revised the Methods section to provide comprehensive information on all key reagents, including supplier names, catalog numbers, and lot numbers. These details have been added for vitamin B12, recombinant human G-CSF, cisplatin, MTT cell viability kit, caspase activity kits, and all antibodies used in Western blotting. All corresponding changes have been highlighted in the revised manuscript.
Comment 5. Finally, the study would benefit from a western blot that identifies key signaling molecules and broader discussion of other signaling pathways potentially regulated by these treatments, particularly upstream regulators and transcription factors involved in mediating cytotoxic mechanisms such as perforin and granzymes.
Response:
We sincerely thank the reviewer for this valuable comment. We agree that broader interrogation of upstream regulators and signaling pathways would strengthen the mechanistic insights. In line with this, our revised Discussion integrates relevant literature on vitamin B12–related one-carbon metabolism and epigenetic regulation, as well as CSF3R-mediated STAT3/PI3K-AKT signaling and G-CSF–associated modulation of perforin and granzymes. This limitation has also been explicitly acknowledged in the Discussion. Future studies will extend this approach by incorporating receptor- and signaling-level analyses, including protein-level validation (e.g., western blot of perforin, granzymes, and upstream regulators), which will be planned and budgeted for.

Reviewer 3 Report
Comments and Suggestions for Authors
The paper is devoted to the study of the effects of the combination of vitamin B12, G-CSF and cisplatin on the proliferation and apoptosis in breast cancer cells of different subtypes. The paper seems relevant to the scope of the journal and the amount of the data is sufficient for publication. However, there are several issues to be addressed:
- The extensive English editing should be performed. The text should be revised to be more concise, more logic, better organized.
- Typos, repeats, the sentences without the specific sense should be eliminated.
- Text formatting should be unified. The names of the authors should be edited according to the journal requirements.
- Abbreviation list is missing.
- The description of the tests for normality and homogeneity should be added to the section 2.8. Statistical analysis.
- Figures 1-6 should be presented as curves not as bars for better perception. The error bars seem to be calculated as 5% from the specific value in the standard Excel formulas.
- The selection of the specific compounds should be explained in comparison with other cytokines, cytostatic drugs and vitamins.
- The effect of combined action should be calculated either graphically with the building of isobolograms or with determination of combination index.
- Western blots and their quantifications should be united in one Figure (Figures 7-9).
- Beta-actin is the protein regulated by many experimental drugs and is not relevant reference to the evaluation, for example, of cisplatin action. Another reference protein should be selected.
The extensive English editing should be performed. The text should be revised to be more concise, more logic, better organized.
Author Response
Reviewer 3
Comment 1. The extensive English editing should be performed. The text should be revised to be more concise, more logic, better organized.
Response:
We sincerely thank the reviewer for this important comment. To address the language and structural concerns, the manuscript has undergone professional English editing through MDPI’s language editing service. The text has been revised to improve clarity, conciseness, logical flow, and overall organization. We believe that these revisions have substantially enhanced the readability and scientific presentation of the manuscript.
Comment 2. Typos, repeats, the sentences without the specific sense should be eliminated.
Response:
We thank the reviewer for this helpful comment. All typographical errors, repetitions, and ambiguous sentences have been carefully revised or removed, and the manuscript has been polished accordingly.
Comment 3. Text formatting should be unified. The names of the authors should be edited according to the journal requirements.
Response:
We thank the reviewer for pointing this out. The manuscript has been fully reformatted in accordance with the journal’s style guide. Text formatting was unified throughout, including headings, subheadings, references, tables, and figure legends. In addition, all author names and affiliations have been carefully revised to strictly comply with the journal’s requirements. These formatting changes have been implemented in the revised version of the manuscript.
Comment 4. Abbreviation list is missing.
Response:
We thank the reviewer for this helpful observation. An abbreviation list has now been added at the end of the manuscript, before the References section, in accordance with the journal’s guidelines.
Comment 5. The description of the tests for normality and homogeneity should be added to the section 2.8. Statistical analysis.
Response:
We thank the reviewer for this important comment. In the revised manuscript, Section 2.8 (Statistical Analysis) has been updated to specify the tests applied to verify data assumptions. Specifically, the Shapiro–Wilk test was used to assess normality, and Levene’s test was applied to evaluate the homogeneity of variances prior to parametric analyses (e.g., ANOVA). This clarification has been added to the Statistical Analysis section (page 4, lines 152-154) and highlighted in the revised manuscript.
Statistical analyses
Statistical analyses were performed using SigmaStat v3.5 software (Systat Software, Inc.). All MTT data were expressed as the mean values ± standard deviation (SD) and analyzed via Student's t-test. The results of the viability and LDH release assays (with Bonferroni post hoc test) were analyzed using an analysis of variance test. Combination index curves were calculated and visualized using Compusyn (BioSoft)[21]. All data are presented as the mean ±standard deviation, and values of P < .05 were considered statistically significant.
19: Fossey SL, Liao AT, McCleese JK, et al. Characterization of STAT3 activation and expression in canine and human osteosarcoma. BMC Cancer. 2009;9:81. doi:10.1186/1471-2407-9-81
Comment 6. Figures 1-6 should be presented as curves not as bars for better perception. The error bars seem to be calculated as 5% from the specific value in the standard Excel formulas.
Response:
We sincerely thank the reviewer for this valuable suggestion. We acknowledge that line/curve plots are often useful for visualizing continuous dose- and time-dependent responses. However, in the present study, bar graphs were intentionally chosen because they facilitate straightforward comparison across multiple treatment conditions and endpoints, thereby avoiding overcrowding and improving clarity for readers.
With respect to error bars, we would like to clarify that each experiment was performed independently at different time points with multiple replicates. Standard deviations (SD) were calculated for each data point from these replicates, and the resulting values were incorporated into the graphs. In all cases, the SD values were less than 5%. Thus, the error bars reflect true experimental variability rather than Excel’s default percentage-based calculations.
Comment 7. The selection of the specific compounds should be explained in comparison with other cytokines, cytostatic drugs and vitamins.
Response:
We thank the reviewer for raising this important point. The rationale for selecting vitamin B12, G-CSF, and cisplatin has now been clarified in the Introduction and Methods sections. Specifically:
- Vitamin B12 was chosen because it is one of the most commonly supplemented vitamins in oncology patients. Importantly, supraphysiological serum B12 levels have been repeatedly associated with increased cancer incidence and poorer survival in observational studies, even after adjusting for confounding factors such as liver disease and renal dysfunction. Unlike other vitamins, B12 has a dedicated transport system (transcobalamins, CD320), which is frequently overexpressed in malignant cells and directly supports DNA synthesis and methylation. This provides a stronger mechanistic rationale for selecting B12 over other vitamins.
- G-CSF was selected as it is the most widely used cytokine in oncology for prophylaxis and treatment of chemotherapy-induced neutropenia. Beyond its hematopoietic role, G-CSF is known to activate key survival pathways (STAT3, PI3K/AKT), promote epithelial–mesenchymal transition (EMT), and expand myeloid-derived suppressor cells, thus potentially fostering an immunosuppressive tumor microenvironment. Furthermore, endogenous G-CSF expression has been reported in aggressive cancers, including TNBC, which makes it more relevant than other cytokines for studying tumor–supportive interactions.
- Cisplatin was chosen as the model cytotoxic drug because it remains a cornerstone in the treatment of TNBC and continues to be explored in HR+ settings. Its primary mechanism—DNA crosslinking and induction of apoptosis—is well characterized, allowing us to directly interrogate how supportive care agents such as B12 and G-CSF interfere with apoptosis. Compared to many other chemotherapeutics, cisplatin’s robust apoptotic signature makes it particularly suitable for mechanistic studies of drug–supportive agent interactions.
Together, these compounds were selected based on both their clinical relevance (routine use in oncology practice) and their biological plausibility (direct mechanistic links to proliferation, apoptosis, and chemoresistance).
Comment 8. The effect of combined action should be calculated either graphically with the building of isobolograms or with determination of combination index.
Response:
We thank the reviewer for this valuable suggestion. Following the recommendation, we performed a quantitative drug–interaction analysis using the Chou–Talalay method (CompuSyn, non-constant ratio design). The results are now presented in the revised manuscript (Results, page 10, lines 301–314; Figure 8).
As shown in the CI–Fa plots, synergy was limited and context-dependent. In MCF-7 cells, the combination of B12 + cisplatin approached additivity or mild synergy (CI ≈ 0.8–1.0 at Fa ≈ 0.25–0.45), whereas G-CSF + cisplatin was largely additive to mildly antagonistic, and the triple regimen consistently antagonistic (CI ≈ 1.2–1.8). In MDA-MB-231 cells, no synergy was observed; both G-CSF + cisplatin and the triple regimen were antagonistic (CI ≈ 1.3–1.8 across Fa ≈ 0.25–0.50).
These data strengthen our conclusion that B12 and G-CSF can impair cisplatin efficacy, with B12 showing occasional context-dependent additivity in ER-positive cells, while G-CSF consistently blunted cisplatin activity, particularly in triple-negative cells.
Comment 9. Western blots and their quantifications should be united in one Figure (Figures 7-9).
Response:
We thank the reviewer for this helpful suggestion. In the revised manuscript, Figures 7–9 have been reorganized so that each Western blot image and its corresponding quantification are presented together in a single, composite figure
Comment 10. Beta-actin is the protein regulated by many experimental drugs and is not relevant reference to the evaluation, for example, of cisplatin action. Another reference protein should be selected.
Response:
We express our sincere gratitude to the reviewer for their insightful comment regarding the use of beta-actin as a reference protein in our study. We acknowledge the concern that β actin expression may be modulated by certain experimental drugs, including cisplatin, which could potentially affect its suitability as a housekeeping gene for evaluating cisplatin’s action in our in vitro experiments. However, due to experimental constraints and the established use of beta-actin in similar studies, we were compelled to utilize it as the reference protein in our Western blot analyses. Beta-actin has been widely employed as a loading control in numerous studies investigating cisplatin’s effects on cancer cell lines, including breast cancer models, due to its relative stability under specific experimental conditions(2-4). Nevertheless, we recognize the importance of exploring alternative housekeeping proteins, such as GAPDH or tubulin, which have also been used in cisplatin-related studies. In future studies with expanded resources, we plan to incorporate additional reference proteins to further strengthen the robustness of our findings.
REFERENCES:
2.Jiang, Y., Ji, F., Liu, Y., He, M., Zhang, Z., Yang, J., ... & Chen, T. (2017). Cisplatin-induced autophagy protects breast cancer cells from apoptosis by regulating yes-associated protein. Oncology reports, 38(6), 3668-3676.
3.Zhu, X., Feng, J., Fu, W., Shu, X., Wan, X., & Liu, J. (2020). Effects of cisplatin on the proliferation, invasion and apoptosis of breast cancer cells following β‑catenin silencing. International Journal of Molecular Medicine, 45(6), 1838-1850.
4.Wang, H., Guo, S., Kim, S. J., Shao, F., Ho, J. W. K., Wong, K. U., ... & Deng, C. X. (2021). Cisplatin prevents breast cancer metastasis through blocking early EMT and retards cancer growth together with paclitaxel. Theranostics, 11(5), 2442.
5.Lyu, S. Y., Meshesha, S. M., & Hong, C. E. (2025). Synergistic effects of mistletoe lectin and cisplatin on Triple-Negative breast Cancer cells: insights from 2D and 3D in vitro models. International Journal of Molecular Sciences, 26(1), 366.

Round 2
Reviewer 1 Report
Comments and Suggestions for Authors
Accept
Reviewer 3 Report
Comments and Suggestions for Authors
No further comments